# Molecular Typing of Mastadenoviruses in Simultaneously Collected Nasopharyngeal Swabs and Stool Samples from Children Hospitalized for Acute Bronchiolitis, Acute Gastroenteritis, and Febrile Seizures

**DOI:** 10.3390/microorganisms11030780

**Published:** 2023-03-17

**Authors:** Urška Glinšek Biškup, Andrej Steyer, Lara Lusa, Franc Strle, Marko Pokorn, Tatjana Mrvič, Štefan Grosek, Miroslav Petrovec, Monika Jevšnik Virant

**Affiliations:** 1Institute of Microbiology and Immunology, Faculty of Medicine, University of Ljubljana, Zaloška 4, 1000 Ljubljana, Slovenia; 2National Laboratory of Health, Environment and Food, Prvomajska 1, 2000 Maribor, Slovenia; 3Department of Mathematics, Faculty of Mathematics, Natural Sciences and Information Technologies, University of Primorska, Glagoljaška 8, 6000 Koper, Slovenia; 4Institute for Biostatistics and Medical Informatics, Faculty of Medicine, University of Ljubljana, Vrazov trg 2, 1104 Ljubljana, Slovenia; 5Department of Infectious Diseases, University Medical Centre Ljubljana, Japljeva 2, 1525 Ljubljana, Slovenia; 6Division of Pediatrics, Ljubljana University Medical Center, Bohoričeva 20, 1000 Ljubljana, Slovenia; 7Department of Paediatrics, Faculty of Medicine, University of Ljubljana, Bohoričeva 20, 1000 Ljubljana, Slovenia; 8Neonatology Section, Department of Perinatology, Division of Obstetrics and Gynecology, University Medical Centre Ljubljana, Šlajmerjeva 3, 1000 Ljubljana, Slovenia; 9Department of Pediatric Intensive Therapy, Division of Pediatrics, University Medical Centre Ljubljana, Bohoričeva 20, 1000 Ljubljana, Slovenia

**Keywords:** mastadenoviruses, acute bronchiolitis, acute gastroenteritis, febrile seizures, clinical study, simultaneously collected nasopharyngeal swabs and stool samples, follow-up, control group

## Abstract

This study determines and compares the frequency of human mastadenovirus (HAdV) presence in children with acute bronchiolitis (AB), acute gastroenteritis (AGE), and febrile seizures (FS), ascertains types of HAdVs associated with each individual syndrome and contrasts the findings with a control group of children. The presence of HAdVs was ascertained in simultaneously collected nasopharyngeal (NP) swabs and stool samples amplifying the hexon gene by RT-PCR; these were sequenced to determine the types of HAdVs. HAdVs were grouped into eight different genotypes. Of these, three (F40, F41, and A31) were found solely in stool samples, whereas the others (B3, C1, C2, C5, and C6) were found in both stool samples and NP swabs. The most common genotypes in NP swabs were C2 (found in children with AGE and FS) and C1 (only in children with FS), whereas in stool samples genotypes F41 (in children with AGE) and C2 (in children with AGE and FS) prevailed, and C2 was simultaneously present in both samples. HAdVs were more often detected in stool samples than in NP swabs in patients (with the highest estimated viral load in stool samples in children with AB and AGE) and healthy controls and were more common in NP swabs in children with AGE than in children with AB. In most patients, the characterized genotypes in NP swabs and stool samples were in concordance.

## 1. Introduction

Human mastadenoviruses (HAdVs) are double-stranded nonenveloped DNA viruses belonging to the family *Adenoviridae* and genus *Mastaendovirus* [1]. Among seven *Human mastadenovirus* species, designated with letters A through G, more than 100 genotypes of HAdVs have been identified (https://talk.ictvonline.org/taxonomy (accessed on 23 January 2023), http://hadvwg.gmu.edu (accessed on 23 January 2023)). The predominant serotypes differ among countries or regions and vary over time [2].

Many HAdV infections are asymptomatic and result in an antibody response that is probably protective against infection with the same serotype [3]. Nevertheless, HAdVs are responsible for a broad spectrum of clinical manifestations, including respiratory infections (predominantly species B, C, and E), keratoconjuctivitis (species D), gastrointestinal manifestations (species A and F), urinary tract infections (species B), and systemic infection in immunocompromised patients (species C) [3,4,5,6]. Genotypes C2, C1, and F41 have mostly been found in infants, and genotypes B3 and B4 mostly in adults [7,8].

It has been estimated that HAdVs cause 5% to 10% of respiratory infections in children and 1% to 7% of respiratory infection in adults [9,10]. Usual symptoms and signs are cough, nasal congestion, and pharyngitis, which cannot be clinically distinguished from streptococcal infection. The respiratory involvement is often accompanied by constitutional symptoms that may be severe. Pneumonia is a common complication of HAdV infection in newborns and infants [4]. The most frequent genotypes found in patients with respiratory infection are C1, C2, C5, and C6, and occasionally genotypes B3 and B7. For lower respiratory tract infection, such as pneumonia, HAdV genotype B7 is usually responsible [7]. Symptomatic infection results in type-specific immunity [11].

In young children, HAdV infections can also cause gastrointestinal symptoms even though their primary site is the respiratory tract [12,13]. The frequency of HAdV infections among children with gastroenteritis is 6% to 14% [14,15,16,17,18,19]. Gastroenteritis is clinically manifested by fever, vomiting, and diarrhea. However, because epidemiologic studies have found HAdVs not only in the stool of patients with diarrhea but also in asymptomatic persons, the role of HAdVs in gastroenteritis is still unclear [19,20]. Interpretation of the presence of HAdVs in stool is complicated due to limited ability to distinguish between asymptomatic shedding of HAdVs and the etiological cause of disease. Nevertheless, two genotypes of HAdVs, F40 and F41, have been commonly associated with diarrhea, whereas several others, so-called non-enteric HAdVs, have been detected in the stool of children with diarrhea at lower frequencies [16,18,21].

Among the many benefits of type-specific identification, including disease prevention strategies, epidemiological investigations, and response to antiviral drugs, molecular characterization may potentially be a basis for more effective treatment in the future [22]. There are many advantages of rapid molecular identification based on polymerase chain reaction (PCR) compared to direct antigen detection: it is rapid, sensitive, and precise. Amplification and sequencing of hexon gene regions, conserved among all recognized HAdV serotypes, have proven to be comparable or better than classic cell culture or immunodiagnostic methods for detection and genotyping of HAdVs in clinical samples [1,13,23,24]. Molecular characterization of HAdV types has been performed by restriction enzyme analysis, multiplex PCR techniques targeting fiber genes and hexon genes, or sequencing of these genes [13]. Molecular typing of HAdVs is based on hypervariable regions of the hexon gene, where antigenic domains have been mapped [1].

This study determines and compares the frequency of HAdV presence in children with acute bronchiolitis (AB), acute gastroenteritis (AGE), and febrile seizures (FS) to determine genotypes of HAdVs associated with each individual syndrome and to compare the findings with those in a control group of healthy children. Moreover, comparison of genetic characteristics of HAdVs in simultaneously collected NP swabs and stool samples of children may contribute to a better understanding of HAdVs in small children.

## 2. Materials and Methods

### 2.1. Study Population

The results presented here were obtained during a 2-year prospective study on viral respiratory and gastrointestinal infections in children under 6 years of age, conducted from October 2009 to September 2011. The study protocol was approved by the National Medical Ethics Committee of the Republic of Slovenia (no. 87/08/09) and was registered at the clinicalTrials.gov registry (reg. NCT00987519). Written informed consent was obtained from the parents of all participants. The principles of the Helsinki Declaration, the Oviedo Convention on Human Rights and Biomedicine, and the Slovenian Code of Medical Deontology were strictly followed in this study.

Children hospitalized with acute bronchiolitis (AB), acute gastroenteritis (AGE), and/or febrile seizures (FS), and a healthy control group, consisting of children without respiratory or gastrointestinal infection, were included in our study. Definitions of diagnosis of hospitalized children included in our study have been described previously [25]. Children with more than one clinical manifestation (i.e., a combination of AB, AGE, and/or FS) were classified under the diagnoses that had been the main reason for hospital admission. A nasopharyngeal (NP) swab and stool sample were obtained upon admission and at a follow-up visit 14 days after initial sampling.

The healthy control group comprised children under 6 years old admitted to the Department of Pediatric Surgery and Intensive Care for elective surgical procedures (mainly for inguinal hernia, testicular retention, and hydrocele testis) during the same time period as the study subjects. In addition to the general rules requiring that only children without infections within the last four weeks prior to surgery be admitted for elective surgical procedures (and that the procedure be postponed in children with symptoms or signs of infection found at examination prior to surgery), this study also obtained additional specific information on the presence of signs and symptoms compatible with gastrointestinal and/or respiratory infection within the previous 14 days. The NP swab and stool sample were obtained from the control group participants upon admission.

### 2.2. Sample Preparation and Nucleic Acid Extraction

Procedures for sample preparation and nucleic acid extraction were performed as described in a previous publication [25].

### 2.3. Detection of Other Viruses

The presence of HAdVs was searched for in NP swabs and stool samples by RT-PCR, amplifying a 132-bp fragment from the hexon gene (Table 1) [26]; a one-step real-time RT-PCR assay was used in a StepOne Real-Time PCR system (Applied Biosystems, Carlsbad, CA). Briefly, 5 μL of total nucleic acid was added to 15 μL of reaction mixture including 2× Reaction Mix, SuperScript^®^ III RT/Platinum^®^ Taq Mix (Invitrogen, Carlsbad, CA, USA) with an additional 6 mM MgSO_4_. The cycling conditions were universal for all respiratory viruses tested: 20 min at 50 °C, 2 min at 95 °C, and 45 cycles of 15 s at 95 °C and 45 s at 60 °C.

In addition, several other viral respiratory agents, including respiratory syncytial virus (RSV) type A and B, human rhinovirus (HRV), human metapneumovirus (HMPV), human coronaviruses (HCoVs), human bocavirus (HBoV), parainfluenza virus (PIV) type 1–3, influenza virus (Flu) type A and B, and enterovirus (EV), were tested in NP swabs [26,27,28,29,30,31,32]. Respiratory viruses were determined by multiplex RT-PCR or single RT-PCR assays, as previously described [25]. Furthermore, several gastrointestinal viral agents were searched for in stool samples in children with AGE and in controls, including rotavirus, astrovirus, norovirus, and coronaviruses [26,31,33,34].

### 2.4. HAdV Sequencing and Characterization

HAdV group-specific primers complementary to regions of the hexon gene conserved among all recognized HAdV serotypes were adopted according to the report of Sriwanna et al. [13]. For the amplification of the 956 bp hexon gene, PCR was performed on a Thermo cycler (Eppendorf, Hamburg, Germany). Briefly, 2 µL of total nucleic acid was added to 23 µL of Platinum^®^ PCR Super Mix (Invitrogen, Carlsbad, CA, USA) and 0.2 µM of each primer (AdV-F2 and AdV-R2; Table 1). PCR products were subsequently purified and sequenced using BigDye terminator chemistry on an ABI PRISM 310 genetic analyzer (Applied Biosystems, Foster City, CA USA).

### 2.5. Statistical Analysis

Numerical variables were summarized with medians and interquartile ranges (IQR), and categorical variables with frequencies and percentages. The distributions of age in the groups were compared using ANOVA. The associations between categorical variables were tested with chi-squared tests, and the distributions of numerical variables were compared using Mann–Whitney tests. Paired tests were used to compare variables from the same patient (stool samples and NP swabs, or samples at baseline and after 14 days). Confidence intervals for percentages were obtained using binomial distribution. All the analyses of Ct-values were restricted to positive patients (because Ct-values were missing for the others). The statistical comparisons between baseline and follow-up positivity results were performed using McNemar’s test.

The distributions of Ct-values within groups were presented graphically using box and whisker plots. The seasonality of positive test results was assessed by estimating the association between the day of the year and test positivity, using periodic restricted cubic splines [35] as implemented in the peRiodiCS R package [36].

We used the level of significance alpha = 0.05. All the analyses were carried out using the R statistical language, version 4.0.2 [37].

## 3. Results

### 3.1. Patients and Controls

Of 867 children included in the study, 717 were acutely ill (307 had AB, 218 AGE, and 192 FS) and 150 constituted the healthy control group. Some of the acutely ill children had more than one clinical syndrome: 21 had AB and AGE, eight had AB and FS, 21 had FS and AGE, and one had clinical indications of all three syndromes.

Children with AB were younger compared to the other two groups (AB: median 11.6 months, IQR: 4.9–19.8; AGE: median 19.5 months, IQR 13.8–28.3; FS: median 18.3 months, IQR 14.7–27.4; *p* < 0.001). Children included in the healthy control group were older than patients (controls: median 25.8 months, IQR 14.8–45.8; patients: median 16.7 months, IQR 9.6–18.3; *p* < 0.001). The female:male ratio was 1:1.3 (313/717; 43.6% females) among patients and 1:6.1 (21/150; 14% females) among controls; differences between the two groups were statistically significant (*p* < 0.001).

### 3.2. Samples

At presentation, 717 NP swabs and 628 stool samples were obtained from 717 children with AB, AGE, or FS, and 150 NP swabs and 150 stool samples were acquired from the healthy control group.

At the follow-up examination 14 days after initial testing, NP swabs were available from 504/717 (70.3%) children, including 236/307 (76.9%) patients that had had AB, 157/218 (72%) patients that had been hospitalized for AGE, and 111/192 (57.8%) patients that had suffered from FS. Stool samples were available at the follow-up examination from 425/628 (67.7%) children, including 180/245 (73.5%) patients that had had AB, 158/218 (72.5%) patients that had had AGE, and 87/165 (52.7%) patients that had suffered from FS. The missing samples (213 NP swabs and 203 stool samples) were from children that did not come to the follow-up examination despite being invited.

### 3.3. HAdV Detection

HAdVs were detected in 106/717 NP swabs (14.8%, 95% CI: 12.3–17.6) and in 188/628 stool samples (29.9%, 95% CI: 26.4–33.7) obtained from patients with AB, AGE, or FS. The corresponding findings in the control group were 14/150 (9.3%, 95% CI: 5.2–15.2) and 40/150 (26.7%, 95% CI: 19.8–34.5) for NP swabs and stool samples, respectively.

Although HAdVs in NP swabs as well as in stool samples were detected more often in patients than in the control group (NP swabs: 14.8% vs. 9.3%, *p* = 0.103; stool samples: 29.9% vs. 26.7%, *p* = 0.490), the differences were not statistically significant.

HAdVs were more often detected in stool samples than in NP swabs in each of the three individual syndromes: AB (26.9% vs. 10.4%), AGE (32.1% vs. 20.2%), and FS (31.5% vs. 15.6%) for stool samples and NP swabs, respectively, as well as in the control group of healthy children (26.7% vs. 9.3%; Table 2).

There was no statistically significant difference in the estimated HAdV load (Ct-value) in NP swabs between patients and controls (median Ct-value 35.6 in patients vs. 34.2; *p* = 0.604), whereas the difference was statistically significant comparing stool samples (median Ct-value 34.4 among patients vs. 37.4 among controls; *p* < 0.001). The estimated viral load in NP swabs was the highest among children with FS (median Ct-value 27.4), statistically significantly higher than in the other two groups of ill children (*p* < 0.001) as well as in comparison to controls (*p* < 0.001). The highest estimated viral load in stool samples was in children with AGE (median Ct-value 32.3), statistically significantly higher than in controls and in the AB group (*p* < 0.001) but not in comparison to the FS group (*p* = 0.364; Figure 1).

HAdVs were found throughout the year with a higher number of positive samples in June in NP swabs and June to September in stool samples, but there was no statistically significant association of seasonality and HAdV positivity (Figure 2 and Figure 3).

Simultaneously detected HAdVs in both NP swabs and stool samples were relatively rare: 78/628 (12.4%) among patients and 11/150 (7.3%) among controls. The prevalence in individual patient groups was 22/245 (9%), 34/218 (15.6%), and 22/165 (13.3%) in children with AB, AGE, and FS, respectively. In patients with AB and AGE and simultaneously detected HAdVs in both samples, estimated viral load was higher in stool samples than in NP swabs (a difference of Ct 4.5, IQR: 1.2 to 8.2 for AB and 5.6, IQR: 0.1 to 18.7 for AGE), whereas in patients with FS the difference was smaller (Ct 1.6 IQR: 1.9 to 3.7; Figure 4 and Figure 5). Overall, there was a positive association between the two Ct-values, which was very weak only in the AGE group.

### 3.4. Presence of Additional Viruses

Within HAdV-positive patients, there was an association between the co-presence of other viruses in NP swabs and the patient group; AB patients had a higher probability compared to the AGE and FS groups (27/32, 84.4%, 95% CI: 67.2% to 94.7%; 27/44, 61.4%, 95% CI: 45.5% to 75.6%; 17/30, 56.7%, 95% CI: 37.4% to 74.5%, in children with AB, AGE, and FS, respectively; *p* = 0.04; Table 3). In the control group, HAdVs were detected with other viruses in 3/14 (21.4%, 95% CI: 4.7% to 50.8%) children.

Co-presence of HAdV with other viruses in stool samples was found in 47/70 (67.1%, 95% CI: 54.9% to 77.9%) patients with AGE; the other two groups (AB and FS) were not tested for the presence of additional gastrointestinal viruses in stool samples (Table 3).

Comparison of HAdV loads in cases with HAdV as the only agent and cases in which HAdVs were co-present with other viruses revealed no significant differences in NP swabs. However, in stool samples of children with AGE, a higher load of HAdV was found in the group with the co-presence of other viruses than in those with a single infection (median Ct-value 26.5 vs. 35.9, respectively).

### 3.5. Follow-Up Testing

Of 504 follow-up NP swabs, 62 (12.3%, 95% CI: 9.6% to 15.5%) were positive for HAdV. The difference in the proportion of HAdV-positive cases comparing baseline and follow-up testing was small and not statistically significant. The majority of positive results (39/62, 62.9%) were positive only at follow-up testing, and only a minority (23/62, 37.1%) were positive at follow-up and also at baseline (*p* = 0.45, McNemar’s test).

In follow-up stool samples, the proportion of HAdV positivity was higher (117/425, 27.5%, 95% CI: 23.3% to 32.0%) than in NP swabs and was comparable to the proportion at baseline (29.9%). The difference between HAdV detected only at follow-up and those detected in both samples (baseline and follow-up) was small (53/117, 45.3% vs. 64/117, 54.7%) and not statistically significant (*p* = 0.84, McNemar’s test).

It seems that HAdV persists longer in stool samples than in NP swabs. Patients that were HAdV-positive in stool samples at initial sampling were more likely HAdV-positive at the follow-up examination (64/114, 56.1%). Figure 6 shows the individual transitions in test results between baseline and the follow-up visit for the patients for whom both measurements were available.

### 3.6. Molecular Characterization and Genotypes of HAdV

Of 527 HAdV-positive DNA isolates from NP swabs and stool samples of patients and controls, at initial testing and at follow-up, 189 had a Ct value ≤ 30 and were thus suitable for sequence analysis. From 189 isolates, 107 sequences were successfully characterized, 72 from stool samples and 35 from NP swabs; of 107 sequenced HAdVs, 105 were from patients and two from healthy control children.

The most common genotypes in NP swabs in patients were C2 (10/26, 38.5%, including five children with AGE and five with FS) and C1 (6/26, 23.1%, including only children with FS). The most common genotypes in stool samples in patients were F41 (19/53, 35.8%, including 15 children with AGE, three with FS, and one with AB) and C2 (13/53, 24.5%, including seven children with FS, four with AGE, and two with AB; Appendix A). The only genotype in the control group was C2, one in a NP swab, and one in a stool sample.

In patients with simultaneously detected HAdVs in NP swabs and stool samples, C2 was the most common genotype. In all but one patient, the characterized genotypes in NP swabs and stool samples were in concordance; in one patient with AGE, HAdV type F41 was characterized in a stool sample and C5 in a NP swab, but the patient had also symptoms characteristic of upper respiratory tract infection (Table 4).

The median Ct value of HAdV type F40 and F41 in stool samples was lower than in HAdV types A, B, and C (16.0 vs. 22.7); the difference was statistically significant (*p* < 0.001).

Partial hexon gene sequence analysis showed little variability within a specific genotype. Despite the short gene nucleotide sequence, there is clear clustering of HAdV genotypes shown on the phylogenetic tree in Figure 7.

## 4. Discussion

HAdVs are associated with a broad spectrum of clinical features, including respiratory tract infections, keratoconjuctivitis, gastrointestinal manifestations, urinary tract infections, and systemic infection [3,4,5,33,34]. This study is the first comparison of genetic characteristics of HAdVs in simultaneously collected NP swabs and stool samples of children with AB, AGE, or FS. Furthermore, to obtain insight into the etiological role of HAdVs in these clinical syndromes, a control group of healthy children was also included and longitudinal follow-up testing was performed 14 days after the initial sampling.

A phylogenetic comparison of 675-nucleotide-long sequences of the hexon gene showed that the HAdVs obtained in this study grouped into eight different genotypes; three of these were found only in stool samples (F40, F41, and A31), whereas the others were present in both sources (B3, C1, C2, C5, and C6). The most common genotypes present in NP swabs were C2 (in children with AGE and FS) and C1 (only in children with FS), representing 61.5% of all genotypes detected in NP swabs. Several studies from Europe show that HAdV types of species C are primarily associated with respiratory infections [38,39,40,41], whereas in Asian countries and Chile the most prevalent species associated with respiratory infections is usually B, followed by species C [8,12,42,43]. It is of interest that, of the two B3 genotypes detected in this study, in both patients (one had AB, the other FS) the genotype was detected simultaneously in a NP swab as well as a stool sample and had comparable Ct-values.

The most common genotypes in stool samples in our patients were F41 (35.5%), predominantly in children with AGE, and C2 (25.5%), detected mostly in children with AGE and FS (Appendix A). These two types have also been reported to be most prevalent in some other studies, but in these studies the proportion of F41 was much higher than C2 [44,45]. Nevertheless, the importance of F41 (and also F40) in gastroenteritis was also found in several other studies [14,16,17,46]. A high proportion of the C2 type, known as a cause of respiratory infection, in stool samples as found in this study could be a consequence of secondarily secreted HAdVs from the respiratory tract into the stool. It has been shown previously that HAdV types infecting the respiratory tract could be shed in the feces over prolonged periods of time [47,48]. In an outbreak of viral gastroenteritis among infants at a daycare center in Japan, the duration of shedding of adenovirus and astrovirus ranged from 1 to 10 days [28]. In this study, C2 was the most common genotype in patients with simultaneously detected HAdVs in NP swabs and stool samples. In all but one patient, the genotypes in the NP swabs and stool samples were in concordance; the exception was a patient with AGE in whom HAdV type F41 was characterized in a stool sample and C5 in a NP swab, but that patient also had symptoms characteristic of upper respiratory tract infection (Appendix A and Table 4).

HAdVs are reported to be the cause of pediatric respiratory infections in approximately 5% to 10% of cases [9,10] and in 6% to 14% of gastrointestinal infections [14,15,16,17,18,19]. Some of the findings of this study were rather surprising and call the etiological role into question. For example, one would expect that NP swabs would be most commonly positive in AB (respiratory infection) and that the highest proportion of HAdVs in stool samples would be in children with AGE. However, this assumption was not valid at all for NP swabs, and it was only partially (formally) applicable to stool samples. Namely, in NP swabs, HAdVs were not most frequently detected in children with AB but in children with AGE (20.2%), followed by children with FS (15.6%), and they were least frequently found in patients with AB (10.4%). Concerning stool samples, although stool samples were most frequently HAdV-positive in children with AGE (32.1%), this proportion was very similar to proportions found in children with AB and FS (26.9% and 31.5%, respectively), as well as in the control group of healthy children (29.9%). Thus, the interpretation of the (etiological) role of HAdVs in children with AGE is difficult and is further complicated by the finding that in 67.1% of children with AGE, in addition to HAdVs, other viruses potentially associated with AGE were present in the stool. A high proportion of HAdV-positive stool samples in asymptomatic children as well as the frequent co-presence of HAdVs and other viruses has been reported previously [14,15,16,17,18,19,20,47,49].

Similar difficulties in interpretation of the etiological role of HAdVs were also valid for children with AB in whom the proportion of HAdV-positive NP swabs was 10.4% vs. 9.3% in the asymptomatic control group. Furthermore, children with AB had a significantly higher probability of having the co-presence of other viruses compared to other groups of children (84.4% vs. 61.4% and 56.7%, in children with AB, AGE, and FS, respectively; *p* = 0.04; Table 3). The co-presence of HAdVs and other viruses has frequently been reported previously [40] and has usually been interpreted to be associated with a more severe illness compared to cases with HAdV detected alone. However, fatal outcomes have also been described in patients in whom solely HAdV was detected [50,51]. In the present study, no difference in illness severity was observed comparing patients with HAdVs detected alone and those with the co-presence of other viruses.

Comparison of the estimated HAdV viral loads (based on Ct-values) in ill children and controls revealed no statistically significant differences for NP swabs (*p* < 0.604), whereas in stool samples the load was significantly higher in patients than in controls (*p* < 0.001). The highest estimated viral load in NP swabs was among children with FS (median Ct-value 27.4, IQR 22.0–34.2) and was significantly higher than in other groups of children (*p* < 0.001) and controls (*p* < 0.001). Children with FS also had a lower proportion of co-presence of other viruses (56.7%) in NP swabs than other groups of children in our study. For a more reliable assessment of the etiological role of HAdVs, HAdV-specific mRNA signaling productive infection needs to be measured [52]. This would explain high Ct-values, exclude false positive DNA detection and ascertain HAdV’s role in co-presence with other viruses.

Although the proportions of children with HAdVs simultaneously detected in both NP swabs and stool samples were higher in patients than in controls, the difference was not statistically significant (78/628, 12.4% vs. 11/150, 7.3%; *p* = 0.106). The proportion was the highest in patients with AGE (34/218, 15.6%), followed by FS (22/165; 13.3%) and AB (22/245; 9%). Nevertheless, the estimated viral load, represented by the median Ct-value, was higher in stool samples than in NP swabs of corresponding patients with AGE (25.9 vs. 34.4) and AB (31.8 vs. 38.5 for AB), but lower in patients with FS (33 vs. 27.4).

Regarding viral load, there has been a lot of discussion of the appropriateness of viral load assessment based on Ct-value. Ct-value is not a direct measurement of viral load. Since it depends upon many host and viral factors, and the laboratory method, it can give just a rough idea of the amount of genomic material in the sample. Wishaupt et al. [53] reported no correlation between HAdV viral load and severity of disease, although interpretation was complicated. In this study, we used viral load (Ct-value) as an indicator of the quantity of viruses in various samples and clinical manifestations with the aim of further assessing the potential etiological role of HAdVs. Nilsen et al. demonstrated that a threshold of 10^6^ copies/mL nasopharyngeal aspirates corresponds to a Ct-value of 30. [54]

It is already known that the epithelial cells of the digestive tract are suitable for HAdV propagation of enteric HAdVs [55,56,57]. However, our results suggest that these cells may also be suitable for non-enteric genotypes of HAdVs.

Insight into the co-presence of HAdVs in two distinct compartments (NP swabs and stool samples) in pediatric patients with AB, AGE, and FS as well as in a control group of healthy children not only made it possible to acquire some simple but interesting results, but also allowed a potential explanation for some of them. The finding that in NP swabs (children with AGE) only non-enteric HAdVs are detected (five C2, two C5, and two C6) whereas in stool samples gastrointestinal as well as non-enteric genotypes were identified (11 F41, two F40, four C2, two C6, one C1, and one C5; Appendix A) corroborates previous knowledge about the suitability of digestive tract epithelial cells for propagation of enteric HAdVs [55,56,57] and also suggests that these cells may also be suitable for non-enteric genotypes of HAdVs. This could be a reasonable explanation for the higher co-presence of HAdVs in NP swabs and stool samples in children with AGE than in patients with AB or FS. It is of interest that the majority (25/33, 75.7%) of children with AGE and co-presence of HAdVs in NP swabs and stool samples also had associated signs of upper respiratory tract infections (Appendix A). This group of children had a higher viral load (lower Ct-value) in stool samples than in NP swabs, suggesting “active” virus in the stool. Our findings are in concordance with a report by Kim et al., in which 119/236 patients (50.4%) with HCoVs detected in stool samples showed concomitant respiratory symptoms, and those with HAdV species C (25/44, 56.8%) showed significantly increased frequencies of respiratory symptoms in comparison to other HAdV species (*p* < 0.01) [44].

In this study, phylogenetic analysis was only performed on a partial hexon gene, which is a more conserved region of the adenoviral genome. Thus, the aim was not to study the divergence of detected strains, but rather to confirm the genotypes detected. There was a clear clustering into HAdV groups A, B, C, and F, and divergence of group C genotypes into four separate phylogenetic branches. However, the latter finding is limited by the small number of strains tested. Further whole-genome investigations may provide additional data on strain variability and consequently the influence on pathogenesis.

## 5. Conclusions

In this study, genetic characteristics of HAdVs in simultaneously collected NP swabs and stool samples of children with AB, AGE, and FS were determined and compared between the three clinical syndromes and a control group of healthy children.

Eight different genotypes were identified: three of them were found only in stool samples (F40, F41, and A31), whereas the others were present in both compartments (B3, C1, C2, C5, and C6). The most commonly detected species in NP swabs was C (92.3%), whereas in stool samples two genotypes predominated: one enteric F41 (35.5%) and one non-enteric C2 (25.5%). The most common genotype in children with simultaneously detected HAdVs in NP swabs and stool samples was C2.

The main practically relevant findings of the present study were: (1) HAdVs were present not only in ill children but also in a healthy control group, (2) HAdVs were more common in NP swabs in children with AGE than in children with AB, and (3) the co-presence of other (non-HAdV) viruses in NP swabs was most common in children with AB, whereas in stool samples this was most common in children with AGE.

Although phylogenetic analysis was performed only on a partial region (but the most hypervariable one) of the hexon gene, this was enough to confirm different HAdV genotypes. For more detailed analysis, further whole-genome investigations would be required.

The etiological role of different HAdV genotypes remains to be determined.

## Figures and Tables

**Figure 1 microorganisms-11-00780-f001:**
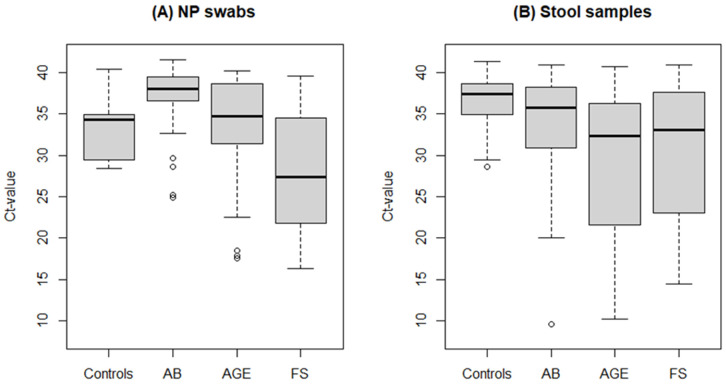
Box and whisker plots of estimated viral load of HAdVs (Ct value) in NP swabs (**A**) and stool samples (**B**) among patients with AB, AGE, and FS, and controls. HAdV = human mastadenovirus, NP = nasopharyngeal swab, AB = acute bronchiolitis, AGE = acute gastroenteritis, FS = febrile seizures, Ct-value = cycle threshold value for participants with positive results, circles represent outlier or single data point.

**Figure 2 microorganisms-11-00780-f002:**
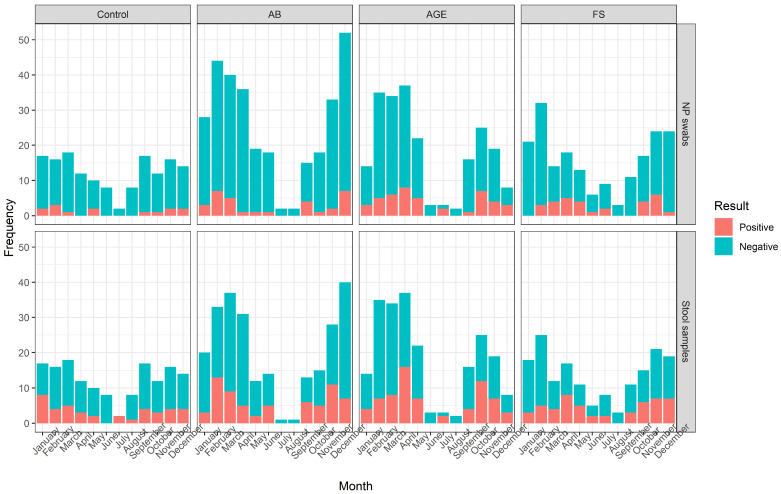
Seasonal distribution of HAdV-positive and -negative results in NP swabs and stool samples, estimated in patients with AB, AGE, and FS, and controls.

**Figure 3 microorganisms-11-00780-f003:**
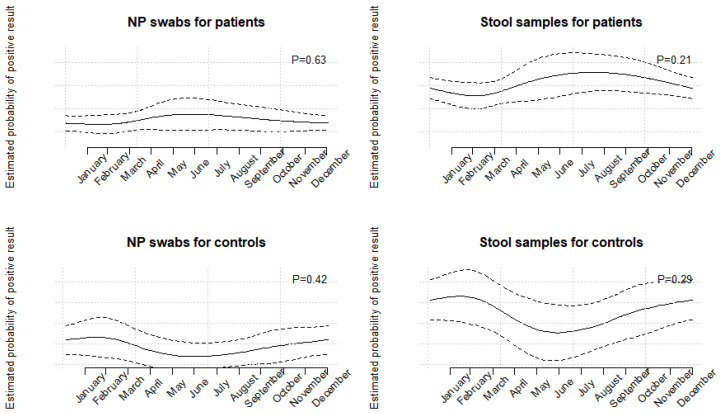
Seasonality of HAdV-positive results in NP swabs and stool samples, estimated separately in patients and controls. Estimates (solid lines) were obtained using periodic restricted cubic splines; dashed lines are pointwise 95% CI. *p*-values test the overall association between time and probability of positive result.

**Figure 4 microorganisms-11-00780-f004:**
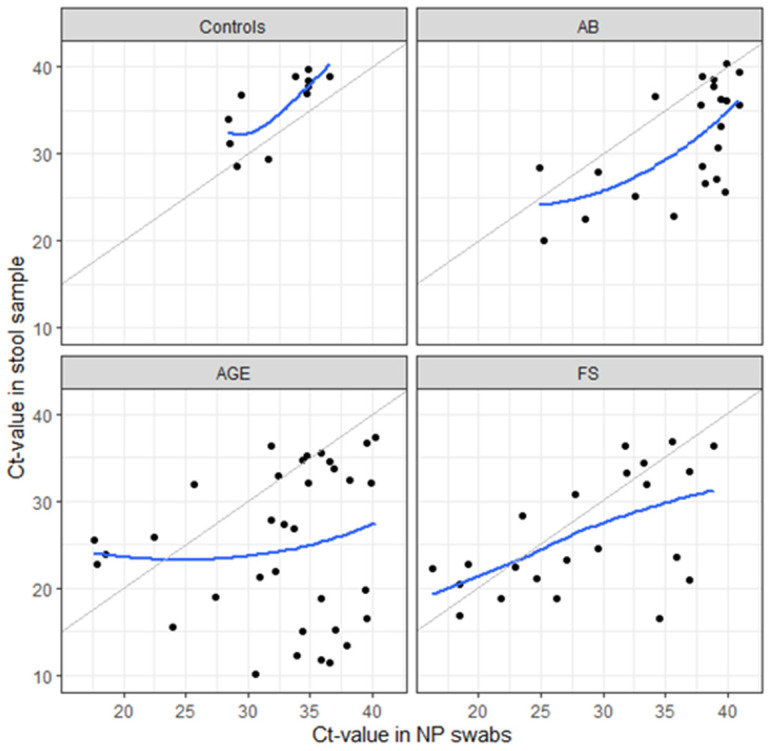
Scatter plots of estimated viral load (Ct-values) of simultaneously detected HAdVs in NP swabs and stool samples in the control and patient groups (AB, AGE, FS). The gray line is the identity line, and the blue line estimates the association between Ct-values in NP swabs using a smooth function.

**Figure 5 microorganisms-11-00780-f005:**
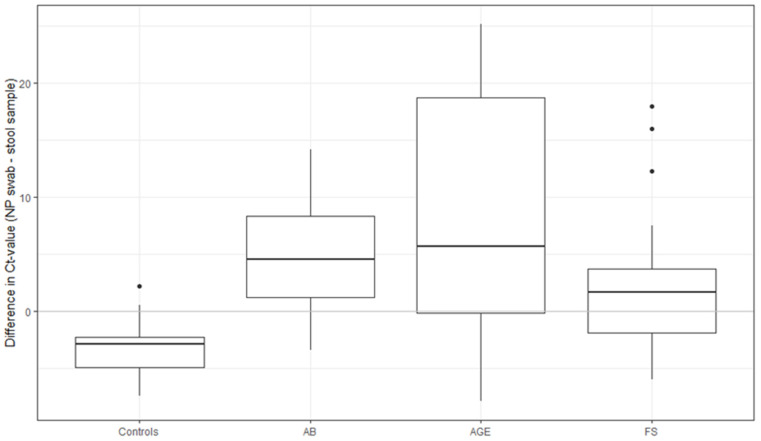
Differences in Ct-value in simultaneously detected HAdVs in both NP swabs and stool samples in the control and patient groups (AB, AGE, FS), circles represent outlier or single data point.

**Figure 6 microorganisms-11-00780-f006:**
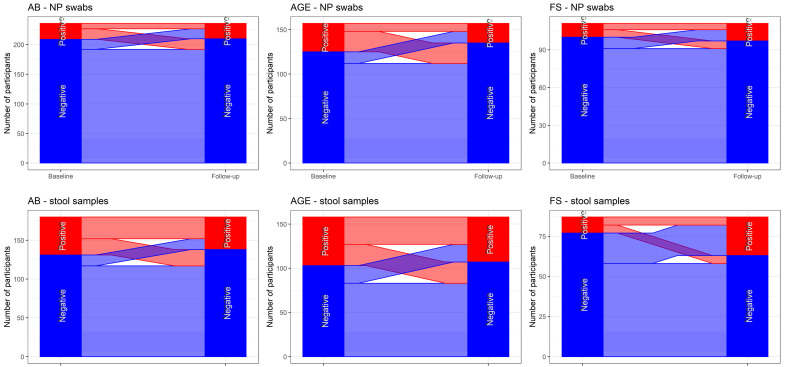
Individual transition in test results between baseline and the follow-up visit for patient groups (AB, AGE, FS).

**Figure 7 microorganisms-11-00780-f007:**
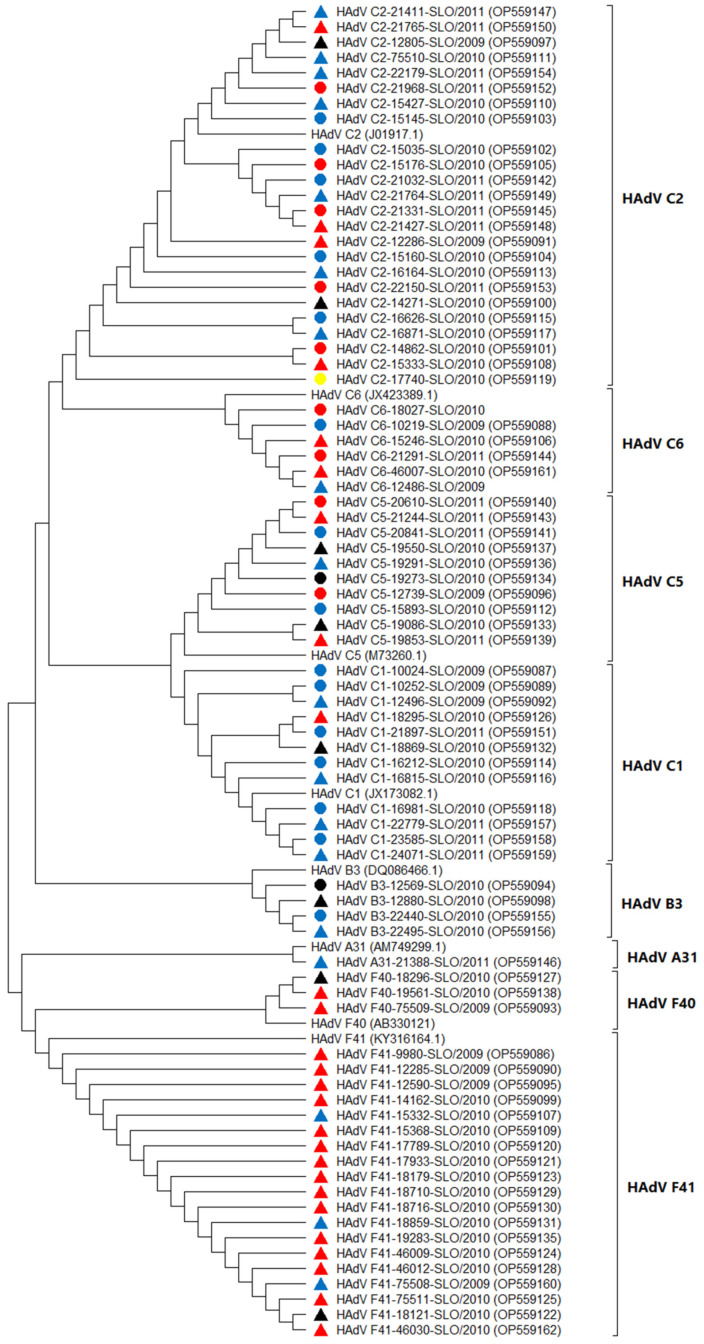
Neighbor-joining phylogenetic tree based on 675-nucleotide-long sequences of a partial hexon gene with 79 representatives of Slovenian HAdV strains detected in nasopharyngeal samples (ο) or stool samples (∆) and eight different HAdV strains from the GeneBank database (with name of HAdV strains and accession numbers). The different colors of the shapes from the Slovenian HAdV strains represent different clinical observations of children: black (AB), red (AGE), blue (FS), and yellow (CO).

**Table 1 microorganisms-11-00780-t001:** Primers for sequence detection of the hexon gene.

Primer/Probe	Sequence, 5′–3′	Detection Technology
AdV-F	GCC ACG GTG GGG TTT CTA AAC TT	RT-PCR
AdV-R	GCC CCA GTG GTC TTA CAT GCA CAT C	RT-PCR
AdV-P-FAM	FAM-TGC ACC AGA CCC GGG CTC AGG TAC TCC GA-TAMRA	RT-PCR
AdV-F2	TTY CCC ATG GCN CAC AAC AC	PCR, sequencing
AdV-R2	GYY TCR ATG AYG CCG CGG TG	PCR, sequencing

**Table 2 microorganisms-11-00780-t002:** HAdV distribution among patients and controls.

Group	AB		AGE		FS		Patients Total		Control Group	
	NP	Stool	NP	Stool	NP	Stool	NP	Stool	NP	Stool
Sample (*n*)	307	245	218	218	192	165	717	628	150	150
HAdV positivity (*n*) % (95% CI)	32 10.4 (7.2–14.4)	66 26.9 (21.5–32.9)	44 20.2 (15.0–26.1)	70 32.1 (25.9–38.7)	30 15.6 (10.8–21.5)	52 31.5 (24.5–39.2)	106 14.8 (12.3–17.6)	188 29.9 (26.4–33.7)	14 9.3 (5.2–15.2)	40 26.7 (19.8–34.5)
Ct-value Median (IQR)	38.1 (36.7–39.5)	35.8 (31.1–38.2)	34.8 (31.7–38.6)	32.3 (21.7–36.3)	27.4 (22.0–34.3)	33 (23.1–37.4)	35.6 (28.3–38.9)	34.4 (25.8–37.1)	34.2 (29.9–34.9)	37.4 (35.0–38.6)

HAdV = human mastadenovirus, NP = nasopharyngeal swab, AB = acute bronchiolitis, AGE = acute gastroenteritis, FS = febrile seizures, CO = control group, Ct-value = cycle threshold value.

**Table 3 microorganisms-11-00780-t003:** Presence of additional viruses in nasopharyngeal swabs or stool samples of patients with AB, AGE, and FS.

	*n* of Positive/*N* Total (%, 95% CI)		
	AB	AGE	FS
NP swabs			
HAdV	32/307 (10.4, 7.2–14.4)	44/218 (20.2, 15.1–26.1)	30/192 (15.6, 10.8–21.5)
HAdV only *	5/32 (15.6, 5.3–32.8)	17/44 (38.6, 24.4–54.5)	13/30 (43.3, 25.5–62.6)
HAdV + additional **	27/32 (84.4, 67.2–94.7)	27/44 (61.4, 45.5–75.6)	17/30 (56.7, 37.4–74.5)
HAdV + 1 ****	20/27 (74.1, 53.7–88.9)	20/27 (74.1, 53.7–88.9)	13/17 (76.5, 50.1–93.2)
HAdV + 2 ****	6/27 (22.2, 8.6–42.3)	6/27 (22.2, 8.6–42.3)	1/17 (5.9, 0.1–28.7)
HAdV + 3 ****	1/27 (3.7, 0.1–19)	1/27 (3.7, 0.1–19)	2/17 (11.8, 1.5–36.4)
HAdV + 4 ****	0/27 (0, 0–12.8)	0/27 (0, 0–12.8)	1/17 (5.9, 0.1–28.7)
INFA ****	0/27 (0, 0–12.8)	0/27 (0, 0–12.8)	2/17 (11.8, 1.5–36.4)
INFB ****	0/27 (0, 0–12.8)	0/27 (0, 0–12.8)	1/17 (5.9, 0.1–28.7)
RSV ****	15/27 (55.5, 35.3–74.5)	7/27 (25.9, 11.1–46.3)	2/17 (11.8, 1.5–36.4)
HCoV ****	4/27 (14.8, 4.2–33.7)	3/27 (11.1, 2.4–29.2)	5/17 (29.4, 10.3–56.0)
HMPV ****	2/27 (7.4. 0.9–24.3)	0/27 (0, 0–12.8)	0/17 (0, 0–19.5)
HBoV ****	4/27 (14.8, 4.2–33.7)	7/27 (25.9, 11.1–46.3)	4/17 (23.5, 6.8–49.9)
HRV ****	10/27 (37, 19.4–57.6)	14/27 (51.8, 31.9–71.3)	5/17 (29.4, 10.3–56.0)
PIV1–3 ****	0/27 (0, 0–12.8)	4/27 (14.8, 4.2–33.7)	6/17 (35.3, 14.2–61.7)
Stool samples			
HAdV		70/218 (32.1, 26.0–38.7)	
HAdV only *		23/70 (32.9, 22.1–45.1)	
HAdV + additional **		47/70 (67.1, 54.9–77.9)	
HAdV + 1 **		41/47 (87.2, 74.3–95.2)	
HAdV + 2 **		6/47 (12.8, 4.8–25.7)	
HAdV + >2 **		0/47 (0, 0–7.5)	
Astro **		1/47 (2.1, 0.1–11.3)	
Noro **		16/47 (34.0, 20.9–49.3)	
Rota **		31/47 (65.9, 50.7–79.1)	
HBoV **		4/47 (8.5, 2.4–20.4)	
HCoVs **		1/47 (2.1, 0.1–11.3)	

AB = acute bronchiolitis, AGE = acute gastroenteritis, FS = febrile seizures, CO = control group of children, HAdV = human mastadenovirus, INF A = influenza virus A, INF B = influenza virus B, RSV = respiratory syncytial virus, HCoV = human coronavirus, HMPV = human metapneumovirus, HBoV = human bocavirus, HRV = human rhinovirus, PIV1–3 = parainfluenza virus 1–3, Astro = astrovirus, Noro = norovirus, Rota = rotavirus, * in HAdV-positives; ** in HAdV-positives with co-infections.

**Table 4 microorganisms-11-00780-t004:** Co-presence of HAdV genotypes in NP swabs and stool samples obtained from children with acute bronchiolitis (AB), acute gastroenteritis (AGE), and febrile seizures (FS).

	Genotypes		
Group	*n*	NP	Stool
AB	1	B3	B3
	1	C5	C5
AGE	1	C5	F41
	3	C2	C2
	1	C5	C5
	1	C6	C6
FS	1	B3	B3
	3	C1	C1
	4	C2	C2
	1	C6	C6
All patients	2	B3	B3
	3	C1	C1
	7	C2	C2
	2	C5	C5
	2	C6	C6
	1	C5	F41

HAdV = human mastadenovirus, AB = acute bronchiolitis, AGE = acute gastroenteritis, FS = febrile seizures, CO = control group, NP = nasopharyngeal swab, Ct-value = cycle threshold value.

## Data Availability

Not applicable.

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
