# Peer review of "Molecular Typing of Mastadenoviruses in Simultaneously Collected Nasopharyngeal Swabs and Stool Samples from Children Hospitalized for Acute Bronchiolitis, Acute Gastroenteritis, and Febrile Seizures"

_microorganisms, 2023, doi:10.3390/microorganisms11030780_

Round 1

Reviewer 1 Report

The article “Molecular typing of mastadenoviruses in simultaneously collected nasopharyngeal swabs and stool samples from children hospitalized for acute bronchiolitis, acute gastroenteritis, and febrile seizures” represents a well-designed original prospective study in a large group of 867 children. Given the high incidence, occasionally – global impact and still incomplete understanding of viral infections, the presented study has high importance. In addition, the findings have significant novelty, being even partially unexpected, as noted by authors. Both the exact data and the related discussion will be of interest for the global scientific community.

I have only few questions and remarks:

1) Considering that the reported data can raise discussions on etiological relations between positive laboratory finding and the disease, I would advise to discuss also the technological background (besides Ct-value) to exclude the hypothetic possibility of false positive laboratory results.

2) Are there any pathogenetic and clinical considerations on the significance of simultaneous positive tests for adeno­viruses and other viruses?

3) Check the formatting of references, including the spacing, please. It should be in accordance with the “Instructions for Authors”.

Finally, I would like to thank the authors for their contribution. It was a pleasure and a true honour to review this manuscript.

Author Response

Monika Jevšnik Virant

Institute of Microbiology and Immunology

Faculty of Medicine

University of Ljubljana

The Editors

Microorganisms

Dear Editors,

Please find enclosed the revised version of the manuscript with the title “Molecular typing of mastadenoviruses in simultaneously collected nasopharyngeal swabs and stool samples from children hospitalized for acute bronchiolitis, acute gastroenteritis, and febrile seizures” and our responses to reviewers' comments.

 We have followed the Instructions for Authors provided and the manuscript has been formatted according to the guidelines.

RESPONSE TO REVIEWERS

Microorganisms-2208029

Reviewer 1

Ad1. Reviewer 1

The article “Molecular typing of mastadenoviruses in simultaneously collected nasopharyngeal swabs and stool samples from children hospitalized for acute bronchiolitis, acute gastroenteritis, and febrile seizures” represents a well-designed original prospective study in a large group of 867 children. Given the high incidence, occasionally – global impact and still incomplete understanding of viral infections, the presented study has high importance. In addition, the findings have significant novelty, being even partially unexpected, as noted by authors. Both the exact data and the related discussion will be of interest for the global scientific community.

Ad1. Authors' response

We would like to thank the Reviewer for this general positive comment about our manuscript.

Ad2. Reviewer 1

I have only few questions and remarks:

Considering that the reported data can raise discussions on etiological relations between positive laboratory finding and the disease, I would advise to discuss also the technological background (besides Ct-value) to exclude the hypothetic possibility of false positive laboratory results.

Ad2. Authors' response

Thank you for your remark. In accordance with Reviewer's suggestion, the following sentences have been included in revised version of the manuscript in the Discussion section: page 19 line 450 to 453 “For a more reliable assessment of the real etiological role of HAdVs, HAdV- specific mRNA signaling productive infection need to be done. These would explain high Ct-values, exclude false positive DNA detection and ascertain HAdVs role in co-presence with other viruses.” 

New reference has been included:

Proenca-Modena, J.L.;de Souza Cardoso R.;Criado M.F.;Milanez G.P.;de Souza W.M.;Parise P.L.;Bertol J.W.;de Jesus B.L.S.;Prates M.C.M.;Silva M.L.;Buzatto G.P.;Demarco R.C.;Valera F.C.P.;Tamashiro E.;Anselmo-Lima W.T.;Arruda E. Human adenovirus replication and persistence in hypertrophic adenoids and palatine tonsils in children. J Med Virol 2019, 91,1250-1262

Ad3. Reviewer 1

Are there any pathogenetic and clinical considerations on the significance of simultaneous positive tests for adeno­viruses and other viruses?

Ad3. Authors' response

We would like to thank the Reviewer for this question. The following sentences have been included in a revised version of the manuscript in the Discussion section: page 19 line 437 to 442: “The co-presence of HAdVs and other viruses has frequently been reported previously [40] and has usually been interpreted to be associated with a more severe illnesses compared to cases with HAdV detected alone. However, fatal outcomes have also been described in patients in whom solely HAdV was detected [50, 51]. In the present study, no difference in illness severity was observed comparing patients with HAdVs detected alone and those with co-presence of other viruses.”

New references have been included:

Probst, V.;Spieker A.J.;Stopczynski T.;Stewart L.S.;Haddadin Z.;Selvarangan R.;Harrison C.J.;Schuster J.E.;Staat M.A.;McNeal M.;Weinberg G.A.;Szilagyi P.G.;Boom J.A.;Sahni L.C.;Piedra P.A.;Englund J.A.;Klein E.J.;Michaels M.G.;Williams J.V.;Campbell A.P.;Patel M.;Gerber S.I.;Halasa N.B. Clinical Presentation and Severity of Adenovirus Detection Alone vs Adenovirus Co-detection With Other Respiratory Viruses in US Children With Acute Respiratory Illness from 2016 to 2018. J Pediatric Infect Dis Soc 2022, 11,430-439

Miyaji, Y.;Kobayashi M.;Sugai K.;Tsukagoshi H.;Niwa S.;Fujitsuka-Nozawa A.;Noda M.;Kozawa K.;Yamazaki F.;Mori M.;Yokota S.;Kimura H. Severity of respiratory signs and symptoms and virus profiles in Japanese children with acute respiratory illness. Microbiol Immunol 2013, 57,811-821

Ad4. Reviewer 1

Check the formatting of references, including the spacing, please. It should be in accordance with the “Instructions for Authors”.

Ad4. Authors' response

All references have been checked and have been formatted in accordance with the “Instructions for Authors”.

Ad5. Reviewer 1

Finally, I would like to thank the authors for their contribution. It was a pleasure and a true honour to review this manuscript.

Ad5. Authors' response

We would like to thank the Reviewer for this very positive comment about our manuscript.

Reviewer 2 Report

In this study (Molecular typing of mastadenoviruses in simultaneously collected nasopharyngeal swabs and stool samples from children hospitalized for acute bronchiolitis, acute gastroenteritis, and febrile seizures) Biškup and colleagues, investigated the presence of HadV in samples from children with acute bronchiolitis (AB), acute gastroenteritis (AGE), and febrile seizures (FS) . They used PCR as a proxy to detect the virus.

The work seems fine, but there are some minor issues such as a lack of details on PCR assay, and the importance of their findings. The conclusions are also poor.

There needs to be a better-defined hypothesis in the introduction. 

They need to say what is the purpose of this study. The methodology is solely based on PCR assay with no other technique that could confirm the presence of viral infection.

Minor concerns:

Please review the abstract it must describe what you did and the main findings

Page 1, Lines 26-28: This sentence is poor and unclear.

Page 3, Line 12: provide details of these individuals included in the control group. Do they were healthy individuals or they were patients with no sign of AGE AB FS?

Page 5, Line 201: Note that, CT-values are not a direct measurement of viral loads. CT can give a rough idea of the amount of genomic material in the sample. For many viruses, CT lower than 20 indicates a high viral burden, and values above 35 may indicate a false positive. You must provide some references that showed the correlation between CT values and HAdV viral loads.

Page 10, line 279: This conclusion is wrong. What you see in fecal samples is just the shedding of viral nucleic acids. You did not perform any assay to recover viable viruses from these samples

Author Response

Monika Jevšnik Virant

Institute of Microbiology and Immunology

Faculty of Medicine

University of Ljubljana

The Editors

Microorganisms

Dear Editors,

Please find enclosed the revised version of the manuscript with the title “Molecular typing of mastadenoviruses in simultaneously collected nasopharyngeal swabs and stool samples from children hospitalized for acute bronchiolitis, acute gastroenteritis, and febrile seizures” and our responses to reviewers' comments.

 We have followed the Instructions for Authors provided and the manuscript has been formatted according to the guidelines.

RESPONSE TO REVIEWERS

Microorganisms-2208029

Reviewer 2

Ad1. Reviewer 2

In this study (Molecular typing of mastadenoviruses in simultaneously collected nasopharyngeal swabs and stool samples from children hospitalized for acute bronchiolitis, acute gastroenteritis, and febrile seizures) Biškup and colleagues, investigated the presence of HadV in samples from children with acute bronchiolitis (AB), acute gastroenteritis (AGE), and febrile seizures (FS). They used PCR as a proxy to detect the virus.

The work seems fine, but there are some minor issues such as a lack of details on PCR assay, and the importance of their findings. The conclusions are also poor.

 Ad1. Authors' response

According to the Reviewer´s suggestion we added information on PCR assay (please see Meterial na Methods, page 3, lines 113 to 138 of the revised article.

The importance of the study has been reported on page 18, line 380 to 384.

Point to point conclusions have been described in the conclusion section of the revised article – please see page 20, lines 509-520.

Ad2. Reviewer 2

There needs to be a better-defined hypothesis in the introduction. 

Ad2. Authors' response

We would like to thank the Reviewer for a good proposal. The following sentence has been included in a revised version of the manuscript in the Introduction section on page 2 line 94: “Moreover, comparison of genetic characteristics of HAdVs in simultaneously collected NP swabs and stool samples of children may contribute to a better understanding of HAdVs in small children.” 

Ad3. Reviewer 2

They need to say what is the purpose of this study. The methodology is solely based on PCR assay with no other technique that could confirm the presence of viral infection.

Ad3. Authors' response

We wold like to thank to Reviewer for his comments.

The purpose of the study is presented in Introduction – please see page 2, line 91 to 96 Unfortunately, the methodology is solely based on PCR assay - we did not use other technique that could confirm the presence of viruses. For better explanation and understanding the following sentences have been included in revised version of the manuscript (please see Discussion section, page 19 line 450 “For a more reliable assessment of the etiological role of HAdVs, HAdV- specific mRNA signaling productive infection need to be done. These would explain high Ct-value, exclude false positive DNA detection and ascertain HAdVs role in co-presence with other viruses.” 

Ad4. Reviewer 2

Minor concerns:

Please review the abstract it must describe what you did and the main findings

Page 1, Lines 26-28: This sentence is poor and unclear.

Ad4. Authors' response

Thank you for your proposal. Please see revised Abstract.

 Ad5. Reviewer 2

Page 3, Line 12: provide details of these individuals included in the control group. Do they were healthy individuals or they were patients with no sign of AGE AB FS?

Ad6. Authors' response

Details provided. Please see page 3 lines 117-118.                        

Ad7. Reviewer 2

Page 5, Line 201: Note that, CT-values are not a direct measurement of viral loads. CT can give a rough idea of the amount of genomic material in the sample. For many viruses, CT lower than 20 indicates a high viral burden, and values above 35 may indicate a false positive. You must provide some references that showed the correlation between CT values and HAdV viral loads.

Ad7. Authors' response 

We agree with Reviewer and expanded Discussion accordingly - please see page 19, line 462 to 469, of the revised article.

New references have been included:

Schjelderup Nilsen, H.J.;Nordbo S.A.;Krokstad S.;Dollner H.;Christensen A. Human adenovirus in nasopharyngeal and blood samples from children with and without respiratory tract infections. J Clin Virol 2019, 111,19-23.

Ad8. Reviewer 2

Page 10, line 279: This conclusion is wrong. What you see in fecal samples is just the shedding of viral nucleic acids. You did not perform any assay to recover viable viruses from these samples

Ad8. Authors' response 

On page 13, line 320 we just explained the observation shown on Figure 6. For better understanding Discussion section has been expanded – please see page 19 line 450 of the revised article:For a more reliable assessment of the etiological role of HAdVs, HAdV- specific mRNAsignaling productive infection need to be done. These would explain high Ct-values, exclude false positive DNA detection and ascertain HAdVs role in co-presence with other viruses.”

Reviewer 3 Report

Authors attempted a molecular survey on the genotypes of mastadenovirus in simultaneously sampled NP swabs and stool samples from children hospitalized for acute bronchiolitis (AB), acute gastroenteritis (AGE), and febrile seizures (FS).

As introduced in lines 49-55, adenovirus can induce a variety of clinical manifestations with more prevalent genotypes, this manuscript did not generate anything new beyond what has already known, except for it reflects the epidemiology in the country in year 2009-2011.

The logics of including FS into this study is not clear.  It did not mean to explore the systemic infection of these children, or for pathogenesis consideration to  link between AB and AGE and FS. There is no explanation on the FS inclusion. Without a  link between these 3 parameters (AB, AGE, FS), plus the similar findings of these virus in the control group, making interpretation difficult and the significance of your findings greatly discounted.

Figures 1-5 can be deleted or placed in supplementary.  Figures are put in the manuscript unless there is something significant. The presentation of Figure 6 is wonderful, but it  adds no value to the conclusion of follow-up study (lines 279-283). Figure 7 can be placed in the supplementary or deleted.

The conclusion is good.  After all, the whole things is boiled down to several key points listed in lines 440-456.

Only Tables and Figure 7 worth to be kept.  I suggest revise into a more concise version (a brief communication) with fewer shower with numbers.

Author Response

Monika Jevšnik Virant

Institute of Microbiology and Immunology

Faculty of Medicine

University of Ljubljana

The Editors

Microorganisms

Dear Editors,

Please find enclosed the revised version of the manuscript with the title “Molecular typing of mastadenoviruses in simultaneously collected nasopharyngeal swabs and stool samples from children hospitalized for acute bronchiolitis, acute gastroenteritis, and febrile seizures” and our responses to reviewers' comments.

 We have followed the Instructions for Authors provided and the manuscript has been formatted according to the guidelines.

RESPONSE TO REVIEWERS

Microorganisms-2208029

Reviewer3

Ad1. Reviewer 3

Authors attempted a molecular survey on the genotypes of mastadenovirus in simultaneously sampled NP swabs and stool samples from children hospitalized for acute bronchiolitis (AB), acute gastroenteritis (AGE), and febrile seizures (FS).

As introduced in lines 49-55, adenovirus can induce a variety of clinical manifestations with more prevalent genotypes, this manuscript did not generate anything new beyond what has already known, except for it reflects the epidemiology in the country in year 2009-2011.

Ad1. Authors' response

We agree with the Reviewer that there is nothing new in technical approaches used in this study however the other approaches are novel. Our study is the first to compare genetic characteristics of HAdVs in simultaneously collected NP swabs and stool samples. It is a big prospective clinical study including detailed clinical information and specimens obtained from children with tree different well-defined clinical entities (AB, AGE, or FS) at the time of acute illness and at follow-up, and a control group.

Ad2. Reviewer 3

The logics of including FS into this study is not clear.  It did not mean to explore the systemic infection of these children, or for pathogenesis consideration to  link between AB and AGE and FS. There is no explanation on the FS inclusion. Without a  link between these 3 parameters (AB, AGE, FS), plus the similar findings of these virus in the control group, making interpretation difficult and the significance of your findings greatly discounted.

Ad2. Authors' response

All three presentations (AB, AGE, or FS) are syndromic clinical entities (i.e. each one has rather distinctive clinical presentation but different causes) and in each of them HAdVs have been implicated as potential causative agents. The aim of the study was to determine and compare the frequency of HAdV presence in children with acute bronchiolitis, acute gastroenteritis, and febrile seizures, to ascertain types of HAdVs associated with each individual syndrome and to contrasts the findings with a control group of children.

Ad3. Reviewer 3

Figures 1-5 can be deleted or placed in supplementary.  Figures are put in the manuscript unless there is something significant. The presentation of Figure 6 is wonderful, but it  adds no value to the conclusion of follow-up study (lines 279-283). Figure 7 can be placed in the supplementary or deleted.

Ad3. Authors' response

Since we think that the message of an article without images will be reduced we would prefer that figures remain a part of the article. However, if the Reviewer insists, we will move the figures to Supplementary material (although it is questionable what significance such a move has for an online journal).

Ad4. Reviewer 3

The conclusion is good.  After all, the whole things is boiled down to several key points listed in lines 440-456.

Ad4 Authors' response

We would like to thank the Reviewer for this positive comment.

Ad5 Reviewer 3

Only Tables and Figure 7 worth to be kept.  I suggest revise into a more concise version (a brief communication) with fewer shower with numbers.

Ad5 Authors' response

Since we think that the message of an article without information depicted on tables will be diminished we would prefer that tables remain a part of the article. Furthermore, Microorganisms publishes three main types of articles: Articles (Original research manuscripts), Reviews and Case reports but not Brief communications with the exception short Communications of preliminary results.

Round 2

Reviewer 3 Report

The V2 of this manuscript show cosmetic improvement.  Still I think the manuscript can be more concise than the current form without affecting the overall findings, but will make it easier to read.  

For example, in Figures 2 and 3, the controls and patient groups (upper and lower portions) can be overlapped (using different colors and symbols) to compare the difference between these two groups.  The backgrounds in the control group are high, so without overlapping them, it is difficult to appreciate their difference from each other.  Similar practice can be applied to other figure if possible.

For another example, authors should try to combine findings of Table 6 and Figure 7.

Much of the, what I think, interesting results, for example, lines 340-344; 536-540 are not reflected in the abstract, after so much analysis was done.  If it is allowed, these information should be incorporated into the abstract.

Author Response

Monika Jevšnik Virant

Institute of Microbiology and Immunology

Faculty of Medicine

University of Ljubljana

The Editors

Microorganisms

Dear Editors,

Please find enclosed the revised version of the manuscript with the title “Molecular typing of mastadenoviruses in simultaneously collected nasopharyngeal swabs and stool samples from children hospitalized for acute bronchiolitis, acute gastroenteritis, and febrile seizures” and our responses to reviewers' comments.

 We have followed the Instructions for Authors provided and the manuscript has been formatted according to the guidelines.

RESPONSE TO REVIEWERS

Microorganisms-2208029

Reviewer 3

Ad1. Reviewer 3

The V2 of this manuscript show cosmetic improvement.  Still I think the manuscript can be more concise than the current form without affecting the overall findings, but will make it easier to read. 

Ad1. Authors's response

According to the Reviewer's suggestions we added missing information in the revised version of the manuscript.

Ad2. Reviewer 3 

For example, in Figures 2 and 3, the controls and patient groups (upper and lower portions) can be overlapped (using different colors and symbols) to compare the difference between these two groups.  The backgrounds in the control group are high, so without overlapping them, it is difficult to appreciate their difference from each other.  Similar practice can be applied to other figure if possible.

Ad2. Authors's response

We would like to thank to Reviewer for this suggestion. Figure 2 shows seasonal distribution and course of the entire research. In Figure 2 we have already used two different colors for positive and negative samples. Combining or overlapping controls and patients would make the images less transparent. This is what we did in Figure 3, where we overlapped patients and controls at the same image with two colors:

We think that this reduces the transparency, not increases it. Therefore, we suggest that Figures 2 and 3 will remain as they are.

Ad3. Reviewer 3

For another example, authors should try to combine findings of Table 6 and Figure 7.

Ad3. Authors'response

We would like to thank to Reviewer for this comment. To combine the findings of Table 6 and Figure 7 is difficult. Table 6 shows the numerical distribution of different HAdV genotypes according to specific clinical manifestations, while Figure 7 shows their phylogenetic relationship and occurrence in different manifestations. We still think that the message of the article without tables and images will be reduced, therefore we move Table 6 to the supplementary material.

Ad4. Reviewer 3

Much of the, what I think, interesting results, for example, lines 340-344; 536-540 are not reflected in the abstract, after so much analysis was done.  If it is allowed, these information should be incorporated into the abstract.

Ad4. Authors'response

Thank you for your proposal. Please see revised Abstract. I must note that the abstract with the added information exceeds the allowed 200 words. If the Editor will agree, then we can exceed the word limit. Unfortunately, the paragraph in lines 536-540 have not be found because our manuscript is only 516 lines long. In the lines 536-540 are references.
